# Riemannian Adaptive Optimization Methods

**Gary Bécigneul, Octavian-Eugen Ganea**
Department of Computer Science
ETH Zürich, Switzerland
{gary.becigneul,octavian.ganea}@inf.ethz.ch

## Abstract

Several first order stochastic optimization methods commonly used in the Euclidean domain such as stochastic gradient descent (SGD), accelerated gradient descent or variance reduced methods have already been adapted to certain Riemannian settings. However, some of the most popular of these optimization tools − namely ADAM, ADAGRAD and the more recent AMSGRAD − remain to be generalized to Riemannian manifolds. We discuss the difficulty of generalizing such adaptive schemes to the most agnostic Riemannian setting, and then provide algorithms and convergence proofs for geodesically convex objectives in the particular case of a product of Riemannian manifolds, in which adaptivity is implemented *across* manifolds in the cartesian product. Our generalization is tight in the sense that choosing the Euclidean space as Riemannian manifold yields the same algorithms and regret bounds as those that were already known for the standard algorithms. Experimentally, we show faster convergence and to a lower train loss value for Riemannian adaptive methods over their corresponding baselines on the realistic task of embedding the WordNet taxonomy in the Poincaré ball.

## 1 Introduction

Developing powerful stochastic gradient-based optimization algorithms is of major importance for a variety of application domains. In particular, for computational efficiency, it is common to opt for a first order method, when the number of parameters to be optimized is great enough. Such cases have recently become ubiquitous in engineering and computational sciences, from the optimization of deep neural networks to learning embeddings over large vocabularies.

This new need resulted in the development of empirically very successful first order methods such as ADAGRAD (Duchi et al., 2011), ADADELTA (Zeiler, 2012), ADAM (Kingma & Ba, 2015) or its recent update AMSGRAD (Reddi et al., 2018).

Note that these algorithms are designed to optimize parameters living in a Euclidean space $\mathbb{R}^n$, which has often been considered as the default geometry to be used for continuous variables. However, a recent line of work has been concerned with the optimization of parameters lying on a *Riemannian manifold*, a more general setting allowing non-Euclidean geometries. This family of algorithms has already found numerous applications, including for instance solving Lyapunov equations (Vandereycken & Vandewalle, 2010), matrix factorization (Tan et al., 2014), geometric programming (Sra & Hosseini, 2015), dictionary learning (Cherian & Sra, 2017) or hyperbolic taxonomy embedding (Nickel & Kiela, 2017; Ganea et al., 2018a; De Sa et al., 2018; Nickel & Kiela, 2018).

A few first order stochastic methods have already been generalized to this setting (see section 6), the seminal one being Riemannian stochastic gradient descent (RSGD) (Bonnabel, 2013), along with new methods for their convergence analysis in the geodesically convex case (Zhang & Sra, 2016). However, the above mentioned empirically successful adaptive methods, together with their convergence analysis, remain to find their respective Riemannian counterparts.

Indeed, the adaptivity of these algorithms can be thought of as assigning one learning rate per coordinate of the parameter vector. However, on a Riemannian manifold, one is generally not given an intrinsic coordinate system, rendering meaningless the notions *sparsity* or *coordinate-wise update*.

**Our contributions.** In this work we *(i)* explain why generalizing these adaptive schemes to the most agnostic Riemannian setting in an intrinsic manner is compromised, and *(ii)* propose generalizations of the algorithms together with their convergence analysis in the particular case of a product of manifolds where each manifold represents one "coordinate" of the adaptive scheme. Finally, we *(iii)* empirically support our claims on the realistic task of hyperbolic taxonomy embedding.

**Our initial motivation.** The particular application that motivated us in developing Riemannian versions of ADAGRAD and ADAM was the learning of symbolic embeddings in non-Euclidean spaces. As an example, the GloVe algorithm (Pennington et al., 2014) − an unsupervised method for learning Euclidean word embeddings capturing semantic/syntactic relationships − benefits significantly from optimizing with ADAGRAD compared to using SGD, presumably because different words are sampled at different frequencies. Hence the absence of Riemannian adaptive algorithms could constitute a significant obstacle to the development of competitive optimization-based Riemannian embedding methods. In particular, we believe that the recent rise of embedding methods in hyperbolic spaces could benefit from such developments (Nickel & Kiela, 2017; 2018; Ganea et al., 2018a;b; De Sa et al., 2018; Vinh et al., 2018).

## 2 PRELIMINARIES AND NOTATIONS

### 2.1 DIFFERENTIAL GEOMETRY

We recall here some elementary notions of differential geometry. For more in-depth expositions, we refer the interested reader to Spivak (1979) and Robbin & Salamon (2011).

**Manifold, tangent space, Riemannian metric.** A *manifold* $\mathcal{M}$ of dimension $n$ is a space that can locally be approximated by a Euclidean space $\mathbb{R}^n$, and which can be understood as a generalization to higher dimensions of the notion of surface. For instance, the sphere $\mathbb{S} := \{x \in \mathbb{R}^n \mid \|x\|_2 = 1\}$ embedded in $\mathbb{R}^n$ is an $(n-1)$-dimensional manifold. In particular, $\mathbb{R}^n$ is a very simple $n$-dimensional manifold, with zero curvature. At each point $x \in \mathcal{M}$, one can define the *tangent space* $T_x\mathcal{M}$, which is an $n$-dimensional vector space and can be seen as a first order local approximation of $\mathcal{M}$ around $x$. A *Riemannian metric* $\rho$ is a collection $\rho := (\rho_x)_{x \in \mathcal{M}}$ of inner-products $\rho_x(\cdot, \cdot) : T_x\mathcal{M} \times T_x\mathcal{M} \to \mathbb{R}$ on $T_x\mathcal{M}$, varying smoothly with $x$. It defines the geometry locally on $\mathcal{M}$. For $x \in \mathcal{M}$ and $u \in T_x\mathcal{M}$, we also write $\|u\|_x := \sqrt{\rho_x(u, u)}$. A *Riemannian manifold* is a pair $(\mathcal{M}, \rho)$.

**Induced distance function, geodesics.** Notice how a choice of a Riemannian metric $\rho$ induces a natural global distance function on $\mathcal{M}$. Indeed, for $x, y \in \mathcal{M}$, we can set $d(x, y)$ to be equal to the infimum of the lengths of smooth paths between $x$ and $y$ in $\mathcal{M}$, where the length $\ell(c)$ of a path $c$ is given by integrating the size of its speed vector $\dot{c}(t) \in T_{c(t)}\mathcal{M}$, in the corresponding tangent space: $\ell(c) := \int_{t=0}^{1} \|\dot{c}(t)\|_{c(t)} dt$. A geodesic $\gamma$ in $(\mathcal{M}, \rho)$ is a smooth curve $\gamma : (a, b) \to \mathcal{M}$ which locally has minimal length. In particular, a shortest path between two points in $\mathcal{M}$ is a geodesic.

**Exponential and logarithmic maps.** Under some assumptions, one can define at point $x \in \mathcal{M}$ the exponential map $\exp_x : T_x\mathcal{M} \to \mathcal{M}$. Intuitively, this map *folds* the tangent space on the manifold. Locally, if $v \in T_x\mathcal{M}$, then for small $t$, $\exp_x(tv)$ tells us how to move in $\mathcal{M}$ as to take a shortest path from $x$ with initial direction $v$. In $\mathbb{R}^n$, $\exp_x(v) = x + v$. In some cases, one can also define the logarithmic map $\log_x : \mathcal{M} \to T_x\mathcal{M}$ as the inverse of $\exp_x$.

**Parallel transport.** In the Euclidean space, if one wants to transport a vector $v$ from $x$ to $y$, one simply translates $v$ along the straight-line from $x$ to $y$. In a Riemannian manifold, the resulting transported vector will depend on which path was taken from $x$ to $y$. The parallel transport $P_x(v; w)$ of a vector $v$ from a point $x$ in the direction $w$ and in a unit time, gives a canonical way to transport $v$ with zero acceleration along a geodesic starting from $x$, with initial velocity $w$.

## 2.2 RIEMANNIAN OPTIMIZATION

Consider performing an SGD update of the form

$$x_{t+1} \leftarrow x_t - \alpha g_t, \tag{1}$$

where $g_t$ denotes the gradient of objective $f_t$[1] and $\alpha > 0$ is the step-size. In a Riemannian manifold $(\mathcal{M}, \rho)$, for smooth $f : \mathcal{M} \rightarrow \mathbb{R}$, Bonnabel (2013) defines Riemannian SGD by the following update:

$$x_{t+1} \leftarrow \exp_{x_t}(-\alpha g_t), \tag{2}$$

where $g_t \in T_{x_t}\mathcal{M}$ denotes the *Riemannian* gradient of $f_t$ at $x_t$. Note that when $(\mathcal{M}, \rho)$ is the Euclidean space $(\mathbb{R}^n, \mathbf{I}_n)$, these two match, since we then have $\exp_x(v) = x + v$.

Intuitively, applying the exponential map enables to perform an update along the shortest path in the relevant direction in unit time, while remaining in the manifold.

In practice, when $\exp_x(v)$ is not known in closed-form, it is common to replace it by a *retraction map* $R_x(v)$, most often chosen as $R_x(v) = x + v$, which is a first-order approximation of $\exp_x(v)$.

## 2.3 AMSGRAD, ADAM, ADAGRAD

Let's recall here the main algorithms that we are taking interest in.

**ADAGRAD.** Introduced by Duchi et al. (2011), the standard form of its update step is defined as[2]

$$x_{t+1}^i \leftarrow x_t^i - \alpha g_t^i / \sqrt{\sum_{k=1}^{t} (g_k^i)^2}. \tag{3}$$

Such updates rescaled coordinate-wise depending on the size of past gradients can yield huge improvements when gradients are sparse, or in deep networks where the size of a good update may depend on the layer. However, the accumulation of *all* past gradients can also slow down learning.

**ADAM.** Proposed by Kingma & Ba (2015), the ADAM update rule is given by

$$x_{t+1}^i \leftarrow x_t^i - \alpha m_t^i / \sqrt{v_t^i}, \tag{4}$$

where $m_t = \beta_1 m_{t-1} + (1 - \beta_1) g_t$ can be seen as a momentum term and $v_t^i = \beta_2 v_{t-1}^i + (1 - \beta_2)(g_t^i)^2$ is an adaptivity term. When $\beta_1 = 0$, one essentially recovers the unpublished method RMSPROP (Tieleman & Hinton, 2012), the only difference to ADAGRAD being that the sum is replaced by an exponential moving average, hence past gradients are forgotten over time in the adaptivity term $v_t$. This circumvents the issue of ADAGRAD that learning could stop too early when the sum of accumulated squared gradients is too significant. Let us also mention that the momentum term introduced by ADAM for $\beta_1 \neq 0$ has been observed to often yield huge empirical improvements.

**AMSGRAD.** More recently, Reddi et al. (2018) identified a mistake in the convergence proof of ADAM. To fix it, they proposed to either modify the ADAM algorithm with[3]

$$x_{t+1}^i \leftarrow x_t^i - \alpha m_t^i / \sqrt{\hat{v}_t^i}, \quad \text{where } \hat{v}_t^i = \max\{\hat{v}_{t-1}^i, v_t^i\}, \tag{5}$$

which they coin AMSGRAD, or to choose an increasing schedule for $\beta_2$, making it time dependent, which they call ADAMNC (for non-constant).

## 3 ADAPTIVE SCHEMES IN RIEMANNIAN MANIFOLDS

### 3.1 THE DIFFICULTY OF DESIGNING ADAPTIVE SCHEMES IN THE GENERAL SETTING

**Intrinsic updates.** It is easily understandable that writing any coordinate-wise update requires the choice of a coordinate system. However, on a Riemannian manifold $(\mathcal{M}, \rho)$, one is generally not

---

[1] to be interpreted as the objective with the same parameters, evaluated at the minibatch taken at time $t$.

[2] a small $\varepsilon = 10^{-8}$ is often added in the square-root for numerical stability, omitted here for simplicity.

[3] with $m_t$ and $v_t$ defined by the same equations as in ADAM (see above paragraph).

provided with a canonical coordinate system. The formalism only allows to work with certain local coordinate systems, also called *charts*, and several different charts can be defined around each point $x \in \mathcal{M}$. One usually says that a quantity defined using a chart is *intrinsic to* $\mathcal{M}$ if its definition *does not depend* on which chart was used. For instance, it is known that the Riemannian gradient $\mathrm{grad} f$ of a smooth function $f : \mathcal{M} \to \mathbb{R}$ can be defined intrinsically to $(\mathcal{M}, \rho)$, but its Hessian is only intrinsically defined at critical points[4]. It is easily seen that the RSGD update of Eq. (2) is intrinsic, since it only involves $\exp$ and $\mathrm{grad}$, which are objects intrinsic to $(\mathcal{M}, \rho)$. However, it is unclear whether it is possible at all to express either of Eqs. (3,4,5) in a coordinate-free or intrinsic manner.

**A tempting solution.** Note that since an update is defined in a tangent space, one could be tempted to fix a canonical coordinate system $e := (e^{(1)}, ..., e^{(n)})$ in the tangent space $T_{x_0}\mathcal{M} \simeq \mathbb{R}^d$ at the initialization $x_0 \in \mathcal{M}$, and parallel-transport $e$ along the optimization trajectory, adapting Eq. (3) to:

$$x_{t+1} \leftarrow \exp_{x_t}(\Delta_t), \quad e_{t+1} \leftarrow P_{x_t}(e_t; \Delta_t), \quad \text{with } \Delta_t := -\alpha g_t \oslash \sqrt{\sum_{k=1}^{t}(g_k)^2}, \quad (6)$$

where $\oslash$ and $(\cdot)^2$ denote coordinate-wise division and square respectively, *these operations being taken relatively to coordinate system* $e_t$. In the Euclidean space, parallel transport between two points $x$ and $y$ does not depend on the path it is taken along because the space has no curvature. However, in a general Riemannian manifold, not only does it depend on the chosen path but curvature will also give to parallel transport a rotational component[5], which will almost surely break the sparsity of the gradients and hence the benefit of adaptivity. Besides, the interpretation of adaptivity as optimizing different features (*i.e.* gradient coordinates) at different speeds is also completely lost here, since the coordinate system used to represent gradients depends on the optimization path. Finally, note that the techniques we used to prove our theorems would not apply to updates defined in the vein of Eq. (6).

## 3.2 Adaptivity is possible across manifolds in a product

From now on, we assume additional structure on $(\mathcal{M}, \rho)$, namely that it is the cartesian product of $n$ Riemannian manifolds $(\mathcal{M}_i, \rho^i)$, where $\rho$ is the induced product metric:

$$\mathcal{M} := \mathcal{M}_1 \times \cdots \times \mathcal{M}_n, \quad \rho := \begin{bmatrix} \rho^1 & & \\ & \ddots & \\ & & \rho^n \end{bmatrix}. \quad (7)$$

**Product notations.** The induced distance function $d$ on $\mathcal{M}$ is known to be given by $d(x, y)^2 = \sum_{i=1}^{n} d^i(x^i, y^i)^2$, where $d^i$ is the distance in $\mathcal{M}_i$. The tangent space at $x = (x^1, ..., x^n)$ is given by $T_x\mathcal{M} = T_{x^1}\mathcal{M}_1 \oplus \cdots \oplus T_{x^n}\mathcal{M}_n$, and the Riemannian gradient $g$ of a smooth function $f : \mathcal{M} \to \mathbb{R}$ at point $x \in \mathcal{M}$ is simply the concatenation $g = ((g^1)^T \cdots (g^n)^T)^T$ of the Riemannian gradients $g^i \in T_{x^i}\mathcal{M}_i$ of each partial map $f^i : y \in \mathcal{M}_i \mapsto f(x^1, ..., x^{i-1}, y, x^{i+1}, ..., x^n)$. Similarly, the exponential, log map and the parallel transport in $\mathcal{M}$ are the concatenations of those in each $\mathcal{M}_i$.

**Riemannian ADAGRAD.** We just saw in the above discussion that designing meaningful adaptive schemes − intuitively corresponding to one learning rate per coordinate − in a general Riemannian manifold was difficult, because of the absence of intrinsic coordinates. Here, we propose to see each component $x^i \in \mathcal{M}^i$ of $x$ as a "coordinate", yielding a simple adaptation of Eq. (3) as

$$x_{t+1}^i \leftarrow \exp_{x_t^i}^i \left( -\alpha g_t^i / \sqrt{\sum_{k=1}^{t} \|g_k^i\|_{x_k^i}^2} \right). \quad (8)$$

**On the adaptivity term.** Note that we take (squared) *Riemannian* norms $\|g_t^i\|_{x_t^i}^2 = \rho_{x_t^i}^i(g_t^i, g_t^i)$ in the adaptivity term rescaling the gradient. In the Euclidean setting, this quantity is simply a scalar $(g_t^i)^2$, which is related to the size of an SGD update of the $i^{th}$ coordinate, rescaled by the learning

---

[4]because the Poisson bracket cancels at critical points (Milnor, 1963, part 1.2).

[5]The rotational component of parallel transport inherited from curvature is called the *holonomy*.

rate (see Eq. (1)): $|g_t^i| = |x_{t+1}^i - x_t^i|/\alpha$. By analogy, note that the size of an RSGD update in $\mathcal{M}_i$ (see Eq. (2)) is given by $d^i(x_{t+1}^i, x_t^i) = d^i(\exp_{x_t^i}^i(-\alpha g_t^i), x_t^i) = \| - \alpha g_t^i\|_{x_t^i}$, hence we also recover $\|g_t^i\|_{x_t^i} = d^i(x_{t+1}^i, x_t^i)/\alpha$, which indeed suggests replacing the scalar $(g_t^i)^2$ by $\|g_t^i\|_{x_t^i}^2$ when transforming a coordinate-wise adaptive scheme into a manifold-wise adaptive one.

## 4 RAMSGRAD, RADAMNC: CONVERGENCE GUARANTEES

In section 2, we briefly presented ADAGRAD, ADAM and AMSGRAD. Intuitively, ADAM can be described as a combination of ADAGRAD with a momentum (of parameter $\beta_1$), with the slight modification that the sum of the past squared-gradients is replaced with an exponential moving average, for an exponent $\beta_2$. Let's also recall that AMSGRAD implements a slight modification of ADAM, allowing to correct its convergence proof. Finally, ADAMNC is simply ADAM, but with a particular non-constant schedule for $\beta_1$ and $\beta_2$. On the other hand, what is interesting to note is that the schedule initially proposed by Reddi et al. (2018) for $\beta_2$ in ADAMNC, namely $\beta_{2t} := 1 - 1/t$, lets $v_t$ recover the sum of squared-gradients of ADAGRAD. Hence, ADAMNC without momentum (*i.e.* $\beta_{1t} = 0$) yields ADAGRAD.

**Assumptions and notations.**  For $1 \leq i \leq n$, we assume $(\mathcal{M}_i, \rho^i)$ is a geodesically complete Riemannian manifold with sectional curvature lower bounded by $\kappa_i \leq 0$. As written in Eq. (7), let $(\mathcal{M}, \rho)$ be the product manifold of the $(\mathcal{M}_i, \rho^i)$'s. For each $i$, let $\mathcal{X}_i \subset \mathcal{M}_i$ be a compact, geodesically convex set and define $\mathcal{X} := \mathcal{X}_1 \times \cdots \times \mathcal{X}_n$, the set of feasible parameters. Define $\Pi_{\mathcal{X}_i} : \mathcal{M}_i \to \mathcal{X}_i$ to be the projection operator, *i.e.* $\Pi_{\mathcal{X}_i}(x)$ is the unique $y \in \mathcal{X}_i$ minimizing $d^i(y, x)$. Denote by $P^i$, $\exp^i$ and $\log^i$ the parallel transport, exponential and log maps in $(\mathcal{M}_i, \rho^i)$, respectively. For $f : \mathcal{M} \to \mathbb{R}$, if $g = \mathrm{grad} f(x)$ for $x \in \mathcal{M}$, denote by $x^i \in \mathcal{M}_i$ and by $g^i \in T_{x^i}\mathcal{M}_i$ the corresponding components of $x$ and $g$. In the sequel, let $(f_t)$ be a family of differentiable, geodesically convex functions from $\mathcal{M}$ to $\mathbb{R}$. Assume that each $\mathcal{X}_i \subset \mathcal{M}_i$ has a diameter bounded by $D_\infty$ and that for all $1 \leq i \leq n, t \in [T]$ and $x \in \mathcal{X}$, $\|(\mathrm{grad} f_t(x))^i\|_{x_i} \leq G_\infty$. Finally, our convergence guarantees will bound the regret, defined at the end of $T$ rounds as $R_T = \sum_{t=1}^T f_t(x_t) - \min_{x \in \mathcal{X}} \sum_{j=1}^T f_j(x)$, so that $R_T = o(T)$. Finally, $\varphi_{x^i \to y^i}^i$ denotes any isometry from $T_{x^i}\mathcal{M}_i$ to $T_{y^i}\mathcal{M}_i$, for $x^i, y^i \in \mathcal{M}_i$.

Following the discussion in section 3.2 and especially Eq. (8), we present Riemannian AMSGRAD in Figure 1a. For comparison, we show next to it the standard AMSGRAD algorithm in Figure 1b.

**Require:** $x_1 \in \mathcal{X}, \{\alpha_t\}_{t=1}^T, \{\beta_{1t}\}_{t=1}^T, \beta_2$
   Set $m_0 = 0, \tau_0 = 0, v_0 = 0$ and $\hat{v}_0 = 0$
   **for** $t = 1$ to $T$ **do** (for all $1 \leq i \leq n$)
      $g_t = \mathrm{grad} f_t(x_t)$
      $m_t^i = \beta_{1t}\tau_{t-1}^i + (1 - \beta_{1t})g_t^i$
      $v_t^i = \beta_2 v_{t-1}^i + (1 - \beta_2)\|g_t^i\|_{x_t^i}^2$
      $\hat{v}_t^i = \max\{\hat{v}_{t-1}^i, v_t^i\}$
      $x_{t+1}^i = \Pi_{\mathcal{X}_i}(\exp_{x_t^i}^i(-\alpha_t m_t^i/\sqrt{\hat{v}_t^i}))$
      $\tau_t^i = \varphi_{x_t^i \to x_{t+1}^i}^i(m_t^i)$
   **end for**

(a) RAMSGRAD in $\mathcal{M}_1 \times \cdots \times \mathcal{M}_n$.

**Require:** $x_1 \in \mathcal{X}, \{\alpha_t\}_{t=1}^T, \{\beta_{1t}\}_{t=1}^T, \beta_2$
   Set $m_0 = 0, v_0 = 0$ and $\hat{v}_0 = 0$
   **for** $t = 1$ to $T$ **do** (for all $1 \leq i \leq n$)
      $g_t = \mathrm{grad} f_t(x_t)$
      $m_t^i = \beta_{1t}m_{t-1}^i + (1 - \beta_{1t})g_t^i$
      $v_t^i = \beta_2 v_{t-1}^i + (1 - \beta_2)(g_t^i)^2$
      $\hat{v}_t^i = \max\{\hat{v}_{t-1}^i, v_t^i\}$
      $x_{t+1}^i = \Pi_{\mathcal{X}_i}(x_t^i - \alpha_t m_t^i/\sqrt{\hat{v}_t^i})$
   **end for**

(b) AMSGRAD in $\mathbb{R}^n$.

Figure 1: Comparison of the Riemannian and Euclidean versions of AMSGRAD.

Write $h_t^i := -\alpha_t m_t^i/\sqrt{\hat{v}_t^i}$. As a natural choice for $\varphi^i$, one could first parallel-transport[6] $m_t^i$ from $x_t^i$ to $\tilde{x}_{t+1}^i := \exp_{x_t^i}(h_t^i)$ using $P^i(\cdot\,; h_t^i)$, and then from $\tilde{x}_{t+1}^i$ to $x_{t+1}^i$ along a minimizing geodesic.

As can be seen, if $(\mathcal{M}_i, \rho_i) = \mathbb{R}$ for all $i$, RAMSGRAD and AMSGRAD coincide: we then have $\kappa_i = 0$, $d^i(x^i, y^i) = |x^i - y^i|$, $\varphi^i = Id$, $\exp_{x^i}^i(v^i) = x^i + v^i$, $\mathcal{M}_1 \times \cdots \times \mathcal{M}_n = \mathbb{R}^n$, $\|g_t^i\|_{x_t^i}^2 = (g_t^i)^2 \in \mathbb{R}$.

---

[6]The idea of parallel-transporting $m_t$ from $T_{x_t}\mathcal{M}$ to $T_{x_{t+1}}\mathcal{M}$ previously appeared in Cho & Lee (2017).

From these algorithms, RADAM and ADAM are obtained simply by removing the $\max$ operations, *i.e.* replacing $\hat{v}_t^i = \max\{\hat{v}_{t-1}^i, v_t^i\}$ with $\hat{v}_t^i = v_t^i$. The convergence guarantee that we obtain for RAMSGRAD is presented in Theorem 1, where the quantity $\zeta$ is defined by Zhang & Sra (2016) as

$$\zeta(\kappa, c) := \frac{c\sqrt{|\kappa|}}{\tanh(c\sqrt{|\kappa|})} = 1 + \frac{c}{3}|\kappa| + \mathcal{O}_{\kappa \to 0}(\kappa^2). \tag{9}$$

For comparison, we also show the convergence guarantee of the original AMSGRAD in appendix C. Note that when $(\mathcal{M}_i, \rho_i) = \mathbb{R}$ for all $i$, convergence guarantees between RAMSGRAD and AMSGRAD coincide as well. Indeed, the curvature dependent quantity $(\zeta(\kappa_i, D_\infty) + 1)/2$ in the Riemannian case then becomes equal to 1, recovering the convergence theorem of AMSGRAD. It is also interesting to understand at which speed does the regret bound worsen when the curvature is small but non-zero: by a multiplicative factor of approximately $1 + D_\infty|\kappa|/6$ (see Eq.(9)). Similar remarks hold for RADAMNC, whose convergence guarantee is shown in Theorem 2. Finally, notice that $\beta_1 := 0$ in Theorem 2 yields a convergence proof for RADAGRAD, whose update rule we defined in Eq. (8).

**Theorem 1** (Convergence of RAMSGRAD). *Let $(x_t)$ and $(\hat{v}_t)$ be the sequences obtained from Algorithm 1a, $\alpha_t = \alpha/\sqrt{t}$, $\beta_1 = \beta_{11}$, $\beta_{1t} \le \beta_1$ for all $t \in [T]$ and $\gamma = \beta_1/\sqrt{\beta_2} < 1$. We then have:*

$$R_T \le \frac{\sqrt{T}D_\infty^2}{2\alpha(1-\beta_1)} \sum_{i=1}^n \sqrt{\hat{v}_T^i} + \frac{D_\infty^2}{2(1-\beta_1)} \sum_{i=1}^n \sum_{t=1}^T \beta_{1t} \frac{\sqrt{\hat{v}_t^i}}{\alpha_t} +$$

$$\frac{\alpha\sqrt{1+\log T}}{(1-\beta_1)^2(1-\gamma)\sqrt{1-\beta_2}} \sum_{i=1}^n \frac{\zeta(\kappa_i, D_\infty) + 1}{2} \sqrt{\sum_{t=1}^T \|g_t^i\|_{x_t^i}^2}. \tag{10}$$

*Proof.* See appendix A. □

**Theorem 2** (Convergence of RADAMNC). *Let $(x_t)$ and $(v_t)$ be the sequences obtained from RADAMNC, $\alpha_t = \alpha/\sqrt{t}$, $\beta_1 = \beta_{11}$, $\beta_{1t} = \beta_1\lambda^{t-1}$, $\lambda < 1$, $\beta_{2t} = 1 - 1/t$. We then have:*

$$R_T \le \sum_{i=1}^n \left( \frac{D_\infty}{2\alpha(1-\beta_1)} + \frac{\alpha(\zeta(\kappa_i, D_\infty) + 1)}{(1-\beta_1)^3} \right) \sqrt{\sum_{t=1}^T \|g_t^i\|_{x_t^i}^2} + \frac{\beta_1 D_\infty^2 G_\infty n}{2\alpha(1-\beta_1)(1-\lambda)^2}. \tag{11}$$

*Proof.* See appendix B. □

**The role of convexity.** Note how the notion of convexity in Theorem 5 got replaced by the notion of geodesic convexity in Theorem 1. Let us compare the two definitions: the differentiable functions $f : \mathbb{R}^n \to \mathbb{R}$ and $g : \mathcal{M} \to \mathbb{R}$ are respectively *convex* and *geodesically convex* if for all $x, y \in \mathbb{R}^n$, $u, v \in \mathcal{M}$:

$$f(x) - f(y) \le \langle \text{grad} f(x), x - y \rangle, \quad g(u) - g(v) \le \rho_u(\text{grad} g(u), -\log_u(v)). \tag{12}$$

But how does this come at play in the proofs? Regret bounds for convex objectives are usually obtained by bounding $\sum_{t=1}^T f_t(x_t) - f_t(x_*)$ using Eq. (12) for any $x_* \in \mathcal{X}$, which boils down to bounding each $\langle g_t, x_t - x_* \rangle$. In the Riemannian case, this term becomes $\rho_{x_t}(g_t, -\log_{x_t}(x_*))$.

**The role of the cosine law.** How does one obtain a bound on $\langle g_t, x_t - x_* \rangle$? For simplicity, let us look at the particular case of an SGD update, from Eq. (1). Using a cosine law, this yields

$$\langle g_t, x_t - x_* \rangle = \frac{1}{2\alpha} \left( \|x_t - x_*\|^2 - \|x_{t+1} - x_*\|^2 \right) + \frac{\alpha}{2}\|g_t\|^2. \tag{13}$$

One now has two terms to bound: *(i)* when summing over $t$, the first one simplifies as a telescopic summation; *(ii)* the second term $\sum_{t=1}^T \alpha_t\|g_t\|^2$ will require a well chosen decreasing schedule for $\alpha$. In Riemannian manifolds, this step is generalized using the analogue lemma 6 introduced by Zhang & Sra (2016), valid in all Alexandrov spaces, which includes our setting of geodesically convex subsets of Riemannian manifolds with lower bounded sectional curvature. The curvature dependent quantity $\zeta$ of Eq. (10) appears from this lemma, letting us bound $\rho_{x_t^i}^i(g_t^i, -\log_{x_t^i}^i(x_*^i))$.

**The benefit of adaptivity.** Let us also mention that the above bounds significantly improve for sparse (per-manifold) gradients. In practice, this could happen for instance for algorithms embedding each word $i$ (or node of a graph) in a manifold $\mathcal{M}_i$ and when just a few words are updated at a time.

**On the choice of $\varphi^i$.** The fact that our convergence theorems (see lemma 3) do not require specifying $\varphi^i$ suggests that the regret bounds could be improved by exploiting momentum/acceleration in the proofs for a particular $\varphi^i$. Note that this remark also applies to AMSGRAD (Reddi et al., 2018).

## 5 EXPERIMENTS

We empirically assess the quality of the proposed algorithms: RADAM, RAMSGRAD and RADAGRAD compared to the non-adaptive RSGD method (Eq. 2). For this, we follow (Nickel & Kiela, 2017) and embed the transitive closure of the WordNet noun hierarchy (Miller et al., 1990) in the $n$-dimensional Poincaré model $\mathbb{D}^n$ of hyperbolic geometry which is well-known to be better suited to embed tree-like graphs than the Euclidean space (Gromov, 1987; De Sa et al., 2018). In this case, each word is embedded in the same space of constant curvature $-1$, thus $\mathcal{M}_i = \mathbb{D}^n, \forall i$. Note that it would also be interesting to explore the benefit of our optimization tools for algorithms proposed in (Nickel & Kiela, 2018; De Sa et al., 2018; Ganea et al., 2018a). The choice of the Poincaré model is justified by the access to closed form expressions for all the quantities used in Alg. 1a:

- Metric tensor: $\rho_x = \lambda_x^2 \mathbf{I}_n, \forall x \in \mathbb{D}^n$, where $\lambda_x = \frac{2}{1-\|x\|^2}$ is the conformal factor.

- Riemannian gradients are rescaled Euclidean gradients: $\mathrm{grad} f(x) = (1/\lambda_x^2) \nabla_E f(x)$.

- Distance function and geodesics, (Nickel & Kiela, 2017; Ungar, 2008; Ganea et al., 2018b).

- Exponential and logarithmic maps: $\exp_x(v) = x \oplus \left( \tanh\left( \frac{\lambda_x \|v\|}{2} \right) \frac{v}{\|v\|} \right)$, where $\oplus$ is the generalized Mobius addition (Ungar, 2008; Ganea et al., 2018b).

- Parallel transport along the unique geodesic from $x$ to $y$: $P_{x \rightarrow y}(v) = \frac{\lambda_x}{\lambda_y} \cdot \mathrm{gyr}[y, -x]v$. This formula was derived from (Ungar, 2008; Ganea et al., 2018b), gyr being given in closed form in (Ungar, 2008, Eq. (1.27)).

**Dataset & Model.** The transitive closure of the WordNet taxonomy graph consists of 82,115 nouns and 743,241 hypernymy Is-A relations (directed edges $\mathcal{E}$). These words are embedded in $\mathbb{D}^n$ such that the distance between words connected by an edge is minimized, while being maximized otherwise. We minimize the same loss function as (Nickel & Kiela, 2017) which is similar with log-likelihood, but approximating the partition function using sampling of negative word pairs (non-edges), fixed to 10 in our case. Note that this loss does not use the direction of the edges in the graph[7]

$$\mathcal{L}(\theta) = \sum_{(u,v) \in \mathcal{E}} \frac{e^{-d_{\mathbb{D}}(u,v)}}{\sum_{u' \in \mathcal{N}(v)} e^{-d_{\mathbb{D}}(u',v)}} \tag{14}$$

**Metrics.** We report both the loss value and the mean average precision (MAP) (Nickel & Kiela, 2017): for each directed edge $(u, v)$, we rank its distance $d(u, v)$ among the full set of ground truth negative examples $\{d(u', v) | (u', v) \notin \mathcal{E}\}$. We use the same two settings as (Nickel & Kiela, 2017), namely: **reconstruction** (measuring representation capacity) and **link prediction** (measuring generalization). For link prediction we sample a validation set of $2\%$ edges from the set of transitive closure edges that contain no leaf node or root. We only focused on 5-dimensional hyperbolic spaces.

**Training details.** For all methods we use the same "burn-in phase" described in (Nickel & Kiela, 2017) for 20 epochs, with a fixed learning rate of 0.03 and using RSGD with retraction as explained in Sec. 2.2. Solely during this phase, we sampled negative words based on their graph degree raised at power 0.75. This strategy improves all metrics. After that, when different optimization methods start, we sample negatives uniformly. We use $n = 5$, following Nickel & Kiela (2017).

---

[7]In a pair $(u, v)$, $u$ denotes the parent, *i.e.* $u$ entails $v$ − parameters $\theta$ are coordinates of all $u, v$.

**Optimization methods.** Experimentally we obtained slightly better results for RADAM over RAMS-GRAD, so we will mostly report the former. Moreover, we unexpectedly observed convergence to lower loss values when replacing the true exponential map with its first order approximation − i.e. the retraction $R_x(v) = x + v$ − in both RSGD and in our adaptive methods from Alg. 1a. One possible explanation is that retraction methods need fewer steps and smaller gradients to "escape" points sub-optimally collapsed on the ball border of $\mathbb{D}^n$ compared to fully Riemannian methods. As a consequence, we report "retraction"-based methods in a separate setting as they are not directly comparable to their fully Riemannian analogues.

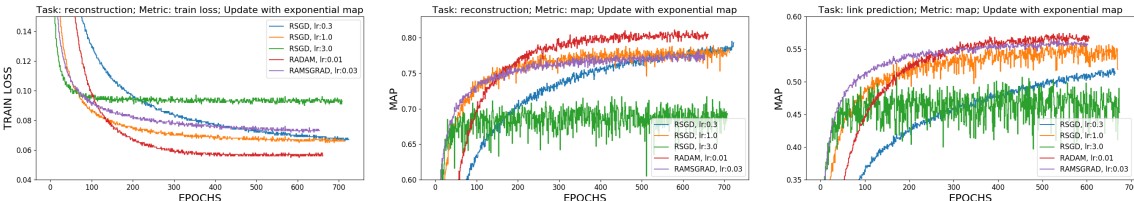

Figure 2: Results for methods doing updates with the exponential map. From left to right we report: training loss, MAP on the train set, MAP on the validation set.

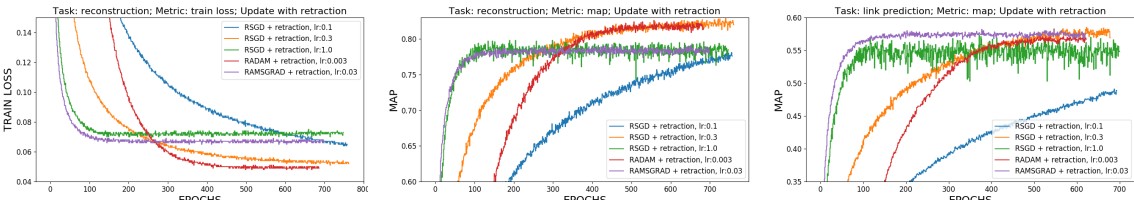

Figure 3: Results for methods doing updates with the retraction. From left to right we report: training loss, MAP on the train set, MAP on the validation set.

**Results.** We show in Figures 2 and 3 results for "exponential" based and "retraction" based methods. We ran all our methods with different learning rates from the set {0.001, 0.003, 0.01, 0.03, 0.1, 0.3, 1.0, 3.0}. For the RSGD baseline we show in orange the best learning rate setting, but we also show the previous lower (slower convergence, in blue) and the next higher (faster overfitting, in green) learning rates. For RADAM and RAMSGRAD we only show the best settings. We always use $\beta_1 = 0.9$ and $\beta_2 = 0.999$ for these methods as these achieved the lowest training loss. RADAGRAD was consistently worse, so we do not report it. As can be seen, RADAM always achieves the lowest training loss. On the MAP metric for both reconstruction and link prediction settings, the same method also outperforms all the other methods for the full Riemannian setting (i.e. Tab. 2). Interestingly, in the "retraction" setting, RADAM reaches the lowest training loss value and is on par with RSGD on the MAP evaluation for both reconstruction and link prediction settings. However, RAMSGRAD is faster to converge in terms of MAP for the link prediction task, suggesting that this method has a better generalization capability.

## 6 RELATED WORK

After Riemannian SGD was introduced by Bonnabel (2013), a plethora of other first order Riemannian methods arose, such as Riemannian SVRG (Zhang et al., 2016), Riemannian Stein variational gradient descent (Liu & Zhu, 2017), Riemannian accelerated gradient descent (Liu et al., 2017; Zhang & Sra, 2018) or averaged RSGD (Tripuraneni et al., 2018), along with new methods for their convergence analysis in the geodesically convex case (Zhang & Sra, 2016). Stochastic gradient Langevin dynamics was generalized as well, to improve optimization on the probability simplex (Patterson & Teh, 2013).

Let us also mention that Roy et al. (2018) proposed Riemannian counterparts of SGD with momentum and RMSprop, suggesting to transport the momentum term using parallel translation, which is an idea that we preserved. However *(i)* no convergence guarantee is provided and *(ii)* their algorithm

performs the coordinate-wise adaptive operations (squaring and division) w.r.t. a coordinate system in the tangent space, which, as we discussed in section 3.1, compromises the possibility of obtaining convergence guarantees. Finally, another version of Riemannian ADAM for the Grassmann manifold $\mathcal{G}(1, n)$ was previously introduced by Cho & Lee (2017), also transporting the momentum term using parallel translation. However, their algorithm completely removes the adaptive component, since the adaptivity term $v_t$ becomes a scalar. No adaptivity across manifolds is discussed, which is the main point of our discussion. Moreover, no convergence analysis is provided either.

## 7 CONCLUSION

Driven by recent work in learning non-Euclidean embeddings for symbolic data, we propose to generalize popular adaptive optimization tools (*e.g.* ADAM, AMSGRAD, ADAGRAD) to Cartesian products of Riemannian manifolds in a principled and intrinsic manner. We derive convergence rates that are similar to the Euclidean corresponding models. Experimentally we show that our methods outperform popular non-adaptive methods such as RSGD on the realistic task of hyperbolic word taxonomy embedding.

ACKNOWLEDGMENTS

Gary Bécigneul is funded by the Max Planck ETH Center for Learning Systems. Octavian Ganea is funded by the Swiss National Science Foundation (SNSF) under grant agreement number 167176.

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

# A  PROOF OF THEOREM 1

*Proof.* Denote by $\tilde{x}_{t+1}^i := \exp_{x_t^i}^i(-\alpha_t m_t^i/\sqrt{\hat{v}_t^i})$ and consider the geodesic triangle defined by $\tilde{x}_{t+1}^i$, $x_t^i$ and $x_*^i$. Now let $a = d^i(\tilde{x}_{t+1}^i, x_*^i)$, $b = d^i(\tilde{x}_{t+1}^i, x_t^i)$, $c = d^i(x_t^i, x_*^i)$ and $A = \angle\tilde{x}_{t+1}^i x_t^i x_*^i$. Combining the following formula[8]:

$$d^i(x_t^i, \tilde{x}_{t+1}^i) d^i(x_t^i, x_*^i) \cos(\angle\tilde{x}_{t+1}^i x_t^i x_*^i) = \langle -\alpha_t m_t^i/\sqrt{\hat{v}_t^i}, \log_{x_t^i}^i(x_*^i)\rangle_{x_t^i}, \tag{15}$$

with the following inequality (given by lemma 6):

$$a^2 \le \zeta(\kappa, c) b^2 + c^2 - 2bc\cos(A), \quad \text{with} \quad \zeta(\kappa, c) := \frac{\sqrt{|\kappa|}c}{\tanh(\sqrt{|\kappa|}c)}, \tag{16}$$

yields

$$\langle -m_t^i, \log_{x_t^i}^i(x_i^*)\rangle_{x^i} \le \frac{\sqrt{\hat{v}_t^i}}{2\alpha_t}\left(d^i(x_t^i, x_*^i)^2 - d^i(\tilde{x}_{t+1}^i, x_*^i)^2\right)$$
$$+ \zeta(\kappa_i, d^i(x_t^i, x_*^i))\frac{\alpha_t}{2\sqrt{\hat{v}_t^i}}\|m_t^i\|_{x_t^i}^2, \tag{17}$$

where the use the notation $\langle\cdot,\cdot\rangle_{x^i}$ for $\rho_{x^i}^i(\cdot,\cdot)$ when it is clear which metric is used. By definition of $\Pi_{\mathcal{X}_i}$, we can safely replace $\tilde{x}_{t+1}^i$ by $x_{t+1}^i$ in the above inequality. Plugging $m_t^i = \beta_{1t}\varphi_{x_{t-1}^i \to x_t^i}^i(m_{t-1}^i) + (1 - \beta_{1t})g_t^i$ into Eq. (17) gives us

$$\langle -g_t^i, \log_{x_t^i}(x_i^*)\rangle_{x_t^i} \le \frac{\sqrt{\hat{v}_t^i}}{2\alpha_t(1 - \beta_{1t})}\left(d^i(x_t^i, x_*^i)^2 - d^i(x_{t+1}^i, x_*^i)^2\right)$$
$$+ \zeta(\kappa_i, d^i(x_t^i, x_*^i))\frac{\alpha_t}{2(1 - \beta_{1t})\sqrt{\hat{v}_t^i}}\|m_t^i\|_{x_t^i}^2$$
$$+ \frac{\beta_{1t}}{(1 - \beta_{1t})}\langle\varphi_{x_{t-1}^i \to x_t^i}^i(m_{t-1}^i), \log_{x_t^i}(x_i^*)\rangle_{x_t^i}. \tag{18}$$

Now applying Cauchy-Schwarz' and Young's inequalities to the last term yields

$$\langle -g_t^i, \log_{x_t^i}(x_i^*)\rangle_{x_t^i} \le \frac{\sqrt{\hat{v}_t^i}}{2\alpha_t(1 - \beta_{1t})}\left(d^i(x_t^i, x_*^i)^2 - d^i(x_{t+1}^i, x_*^i)^2\right)$$
$$+ \zeta(\kappa_i, d^i(x_t^i, x_*^i))\frac{\alpha_t}{2(1 - \beta_{1t})\sqrt{\hat{v}_t^i}}\|m_t^i\|_{x_t^i}^2$$
$$+ \frac{\beta_{1t}}{2(1 - \beta_{1t})}\frac{\alpha_t}{\sqrt{\hat{v}_t^i}}\|m_{t-1}^i\|_{x_{t-1}^i}^2 + \frac{\beta_{1t}}{2(1 - \beta_{1t})}\frac{\sqrt{\hat{v}_t^i}}{\alpha_t}\|\log_{x_t^i}(x_i^*)\|_{x_t^i}^2. \tag{19}$$

From the geodesic convexity of $f_t$ for $1 \le t \le T$, we have

$$\sum_{t=1}^T f_t(x_t) - f_t(x_*) \le \sum_{t=1}^T \langle -g_t, \log_{x_t}(x_*)\rangle_{x_t} = \sum_{i=1}^n \sum_{t=1}^T \langle -g_t^i, \log_{x_t^i}^i(x_*^i)\rangle_{x_t^i}. \tag{20}$$

---

[8]Note that since each $\mathcal{X}_i$ is geodesically convex, logarithms are well-defined.

Let's look at the first term. Using $\beta_{1t} \leq \beta_1$ and with a change of indices, we have

$$\sum_{i=1}^{n}\sum_{t=1}^{T} \frac{\sqrt{\hat{v}_t^i}}{2\alpha_t(1-\beta_{1t})} \left( d^i(x_t^i, x_*^i)^2 - d^i(x_{t+1}^i, x_*^i)^2 \right) \tag{21}$$

$$\leq \frac{1}{2(1-\beta_1)} \left[ \sum_{i=1}^{n}\sum_{t=2}^{T} \left( \frac{\sqrt{\hat{v}_t^i}}{\alpha_t} - \frac{\sqrt{\hat{v}_{t-1}^i}}{\alpha_{t-1}} \right) d^i(x_t^i, x_*^i)^2 + \sum_{i=1}^{n} \frac{\sqrt{\hat{v}_1^i}}{\alpha_1} d^i(x_1^i, x_*^i)^2 \right] \tag{22}$$

$$\leq \frac{1}{2(1-\beta_1)} \left[ \sum_{i=1}^{n}\sum_{t=2}^{T} \left( \frac{\sqrt{\hat{v}_t^i}}{\alpha_t} - \frac{\sqrt{\hat{v}_{t-1}^i}}{\alpha_{t-1}} \right) D_\infty^2 + \sum_{i=1}^{n} \frac{\sqrt{\hat{v}_1^i}}{\alpha_1} D_\infty^2 \right] \tag{23}$$

$$= \frac{D_\infty^2}{2\alpha_T(1-\beta_1)} \sum_{i=1}^{n} \sqrt{\hat{v}_T^i}, \tag{24}$$

where the last equality comes from a standard telescopic summation. We now need the following lemma.

**Lemma 3.**

$$\sum_{t=1}^{T} \frac{\alpha_t}{\sqrt{\hat{v}_t^i}} \|m_t^i\|_{x_t^i}^2 \leq \frac{\alpha\sqrt{1+\log T}}{(1-\beta_1)(1-\gamma)\sqrt{1-\beta_2}} \sqrt{\sum_{t=1}^{T} \|g_t^i\|_{x_t^i}^2} \tag{25}$$

*Proof.* Let's start by separating the last term, and removing the hat on $v$.

$$\sum_{t=1}^{T} \frac{\alpha_t}{\sqrt{\hat{v}_t^i}} \|m_t^i\|_{x_t^i}^2 \leq \sum_{t=1}^{T-1} \frac{\alpha_t}{\sqrt{\hat{v}_t^i}} \|m_t^i\|_{x_t^i}^2 + \frac{\alpha_T}{\sqrt{\hat{v}_T^i}} \|m_T^i\|_{x_T^i}^2 \tag{26}$$

$$\leq \sum_{t=1}^{T-1} \frac{\alpha_t}{\sqrt{\hat{v}_t^i}} \|m_t^i\|_{x_t^i}^2 + \frac{\alpha_T}{\sqrt{v_T^i}} \|m_T^i\|_{x_T^i}^2 \tag{27}$$

Let's now have a closer look at the last term. We can reformulate $m_T^i$ as:

$$m_T^i = \sum_{j=1}^{T}(1-\beta_{1j})\left(\prod_{k=1}^{T-j}\beta_{1,(T-k+1)}\right) \varphi_{x_{T-1}^i \to x_T^i}^i \circ \cdots \circ \varphi_{x_j^i \to x_{j+1}^i}^i (g_j^i) \tag{28}$$

Applying lemma 7, we get

$$\|m_T^i\|_{x_T^i}^2 \leq \left(\sum_{j=1}^{T}(1-\beta_{1j})\left(\prod_{k=1}^{T-j}\beta_{1,(T-k+1)}\right)\right) \times$$

$$\left(\sum_{j=1}^{T}(1-\beta_{1j})\left(\prod_{k=1}^{T-j}\beta_{1,(T-k+1)}\right) \|\varphi_{x_{T-1}^i \to x_T^i}^i \circ \cdots \circ \varphi_{x_j^i \to x_{j+1}^i}^i (g_j^i)\|_{x_T^i}^2\right). \tag{29}$$

Since $\varphi^i$ is an isometry, we always have $\|\varphi_{x \to y}^i(u)\|_y = \|u\|_x$, *i.e.*

$$\|\varphi_{x_{T-1}^i \to x_T^i}^i \circ \cdots \circ \varphi_{x_j^i \to x_{j+1}^i}^i (g_j^i)\|_{x_T^i}^2 = \|g_j^i\|_{x_j^i}^2. \tag{30}$$

Using that $\beta_{1k} \leq \beta_1$ for all $k \in [T]$,

$$\|m_T^i\|_{x_T^i}^2 \leq \left(\sum_{j=1}^{T}(1-\beta_{1j})\beta_1^{T-j}\right)\left(\sum_{j=1}^{T}(1-\beta_{1j})\beta_1^{T-j}\|g_j^i\|_{x_j^i}^2\right). \tag{31}$$

Finally, $(1-\beta_{1j}) \leq 1$ and $\sum_{j=1}^{T}\beta_1^{T-j} \leq 1/(1-\beta_1)$ yield

$$\|m_T^i\|_{x_T^i}^2 \leq \frac{1}{1-\beta_1}\left(\sum_{j=1}^{T}\beta_1^{T-j}\|g_j^i\|_{x_j^i}^2\right). \tag{32}$$

Let's now look at $v_T^i$. It is given by

$$v_T^i = (1 - \beta_2) \sum_{j=1}^{T} \beta_2^{T-j} \|g_j^i\|_{x_j^i}^2 \tag{33}$$

Combining Eq. (32) and Eq. (33) allows us to bound the last term of Eq. (26):

$$\frac{\alpha_T}{\sqrt{v_T^i}} \|m_T^i\|_{x_T^i}^2 \leq \frac{\alpha}{(1-\beta_1)\sqrt{T}} \frac{\left(\sum_{j=1}^{T} \beta_1^{T-j} \|g_j^i\|_{x_j^i}^2\right)}{\sqrt{(1-\beta_2) \sum_{j=1}^{T} \beta_2^{T-j} \|g_j^i\|_{x_j^i}^2}} \tag{34}$$

$$\leq \frac{\alpha}{(1-\beta_1)\sqrt{T}} \sum_{j=1}^{T} \frac{\left(\beta_1^{T-j} \|g_j^i\|_{x_j^i}^2\right)}{\sqrt{(1-\beta_2)\beta_2^{T-j} \|g_j^i\|_{x_j^i}^2}} \tag{35}$$

$$= \frac{\alpha}{(1-\beta_1)\sqrt{T(1-\beta_2)}} \sum_{j=1}^{T} \gamma^{T-j} \|g_j^i\|_{x_j^i} \tag{36}$$

With this inequality, we can now bound every term of Eq. (26):

$$\sum_{t=1}^{T} \frac{\alpha_t}{\sqrt{\hat{v}_t^i}} \|m_t^i\|_{x_t^i}^2 \leq \sum_{t=1}^{T} \frac{\alpha}{(1-\beta_1)\sqrt{t(1-\beta_2)}} \sum_{j=1}^{t} \gamma^{t-j} \|g_j^i\|_{x_j^i} \tag{37}$$

$$= \frac{\alpha}{(1-\beta_1)\sqrt{1-\beta_2}} \sum_{t=1}^{T} \frac{1}{\sqrt{t}} \sum_{j=1}^{t} \gamma^{t-j} \|g_j^i\|_{x_j^i} \tag{38}$$

$$= \frac{\alpha}{(1-\beta_1)\sqrt{1-\beta_2}} \sum_{t=1}^{T} \|g_t^i\|_{x_j^i} \sum_{j=t}^{T} \gamma^{j-t}/\sqrt{j} \tag{39}$$

$$\leq \frac{\alpha}{(1-\beta_1)\sqrt{1-\beta_2}} \sum_{t=1}^{T} \|g_t^i\|_{x_j^i} \sum_{j=t}^{T} \gamma^{j-t}/\sqrt{t} \tag{40}$$

$$\leq \frac{\alpha}{(1-\beta_1)\sqrt{1-\beta_2}} \sum_{t=1}^{T} \|g_t^i\|_{x_j^i} \frac{1}{(1-\gamma)\sqrt{t}} \tag{41}$$

$$\leq \frac{\alpha}{(1-\beta_1)(1-\gamma)\sqrt{1-\beta_2}} \sqrt{\sum_{t=1}^{T} \|g_t^i\|_{x_j^i}^2} \sqrt{\sum_{t=1}^{T} \frac{1}{t}} \tag{42}$$

$$\leq \frac{\alpha\sqrt{1+\log T}}{(1-\beta_1)(1-\gamma)\sqrt{1-\beta_2}} \sqrt{\sum_{t=1}^{T} \|g_t^i\|_{x_j^i}^2} \tag{43}$$

$$\square$$

Putting together Eqs. (19), (20), (24) and lemma 3 lets us bound the regret:

$$\sum_{t=1}^{T} f_t(x_t) - f_t(x_*) \leq \sum_{i=1}^{n} \sum_{t=1}^{T} \langle -g_t^i, \log_{x_t^i}(x_i^*) \rangle_{x_t^i} \tag{44}$$

$$\leq \frac{\sqrt{T}D_\infty^2}{2\alpha(1-\beta_1)} \sum_{i=1}^{n} \sqrt{\hat{v}_T^i} + \frac{D_\infty^2}{2(1-\beta_1)} \sum_{i=1}^{n} \sum_{t=1}^{T} \beta_{1t} \frac{\sqrt{\hat{v}_t^i}}{\alpha_t} \tag{45}$$

$$+ \frac{\alpha\sqrt{1+\log T}}{(1-\beta_1)^2(1-\gamma)\sqrt{1-\beta_2}} \sum_{i=1}^{n} \frac{\zeta(\kappa_i, D_\infty) + 1}{2} \sqrt{\sum_{t=1}^{T} \|g_t^i\|_{x_j^i}^2}, \tag{46}$$

where we used the facts that $d \mapsto \zeta(\kappa, d)$ is an increasing function, and that $\alpha_t / \sqrt{\hat{v}_t^i} \leq \alpha_{t-1} / \sqrt{\hat{v}_{t-1}^i}$, which enables us to bound both the second and third terms of the right-hand side of Eq. (19) using lemma 3. □

**Remark.** Let us notice that similarly as for AMSGRAD, RAMSGRAD also has a regret bounded by $\mathcal{O}(G_\infty \sqrt{T})$. This is easy to see from the proof of lemma 4. Hence the actual upper-bound on the regret is a minimum between the one in $\mathcal{O}(G_\infty \sqrt{T})$ and the one of Theorem 1.

## B PROOF OF THEOREM 2

*Proof.* Similarly as for the proof of Theorem 1 (and with same notations), we obtain the inequality:

$$
\begin{aligned}
\langle -g_t^i, \log_{x_t^i}(x_i^*) \rangle_{x_t^i} \leq &\frac{\sqrt{v_t^i}}{2\alpha_t(1 - \beta_{1t})} \left( d^i(x_t^i, x_*^i)^2 - d^i(x_{t+1}^i, x_*^i)^2 \right) \\
&+ \zeta(\kappa_i, d^i(x_t^i, x_*^i)) \frac{\alpha_t}{2(1 - \beta_{1t})\sqrt{v_t^i}} \|m_t^i\|_{x_t^i}^2 \\
&+ \frac{\beta_{1t}}{2(1 - \beta_{1t})} \frac{\alpha_t}{\sqrt{v_t^i}} \|m_{t-1}^i\|_{x_{t-1}^i}^2 + \frac{\beta_{1t}}{2(1 - \beta_{1t})} \frac{\sqrt{v_t^i}}{\alpha_t} \|\log_{x_t^i}(x_i^*)\|_{x_t^i}^2. \quad (47)
\end{aligned}
$$

From the geodesic convexity of $f_t$ for $1 \leq t \leq T$, we have

$$
\sum_{t=1}^T f_t(x_t) - f_t(x_*) \leq \sum_{t=1}^T \langle -g_t, \log_{x_t}(x_*) \rangle_{x_t} = \sum_{i=1}^n \sum_{t=1}^T \langle -g_t^i, \log_{x_t^i}(x_*^i) \rangle_{x_t^i}. \quad (48)
$$

With the same techniques as before, we obtain the same bound on the first term:

$$
\sum_{i=1}^n \sum_{t=1}^T \frac{\sqrt{v_t^i}}{2\alpha_t(1 - \beta_{1t})} \left( d^i(x_t^i, x_*^i)^2 - d^i(x_{t+1}^i, x_*^i)^2 \right) \leq \frac{D_\infty^2}{2\alpha_T(1 - \beta_1)} \sum_{i=1}^n \sqrt{v_T^i}. \quad (49)
$$

However, for the other terms, we now need a new lemma:

**Lemma 4.**

$$
\sum_{t=1}^T \frac{\alpha_t}{\sqrt{\hat{v}_t^i}} \|m_t^i\|_{x_t^i}^2 \leq \frac{2\alpha}{(1 - \beta_1)^2} \sqrt{\sum_{t=1}^T \|g_t^i\|_{x_t^i}^2}. \quad (50)
$$

*Proof.* Let's start by separating the last term.

$$
\sum_{t=1}^T \frac{\alpha_t}{\sqrt{v_t^i}} \|m_t^i\|_{x_t^i}^2 \leq \sum_{t=1}^{T-1} \frac{\alpha_t}{\sqrt{v_t^i}} \|m_t^i\|_{x_t^i}^2 + \frac{\alpha_T}{\sqrt{v_T^i}} \|m_T^i\|_{x_T^i}^2. \quad (51)
$$

Similarly as before, we have

$$
\|m_T^i\|_{x_T^i}^2 \leq \frac{1}{1 - \beta_1} \left( \sum_{j=1}^T \beta_1^{T-j} \|g_j^i\|_{x_j^i}^2 \right). \quad (52)
$$

Let's now look at $v_T^i$. Since $\beta_{2t} = 1 - 1/t$, it is simply given by

$$
v_T^i = \sum_{t=1}^T \|g_t^i\|_{x_t^i}^2 / T. \quad (53)
$$

Combining these yields:

$$
\frac{\alpha_T}{\sqrt{v_T^i}} \|m_T^i\|_{x_T^i}^2 \leq \frac{\alpha}{1 - \beta_1} \frac{\sum_{j=1}^T \beta_1^{T-j} \|g_j^i\|_{x_j^i}^2}{\sqrt{\sum_{t=1}^T \|g_t^i\|_{x_t^i}^2}} \leq \frac{\alpha}{1 - \beta_1} \sum_{j=1}^T \frac{\beta_1^{T-j} \|g_j^i\|_{x_j^i}^2}{\sqrt{\sum_{k=1}^j \|g_k^i\|_{x_k^i}^2}}. \quad (54)
$$

Using this inequality at all time-steps gives

$$\sum_{t=1}^{T} \frac{\alpha_t}{\sqrt{v_t^i}} \|m_t^i\|_{x_t^i}^2 \leq \frac{\alpha}{1-\beta_1} \sum_{j=1}^{T} \frac{\sum_{l=0}^{T-j} \beta_1^l \|g_j^i\|_{x_j^i}^2}{\sqrt{\sum_{k=1}^{j} \|g_k^i\|_{x_k^i}^2}} \tag{55}$$

$$\leq \frac{\alpha}{(1-\beta_1)^2} \sum_{j=1}^{T} \frac{\|g_j^i\|_{x_j^i}^2}{\sqrt{\sum_{k=1}^{j} \|g_k^i\|_{x_k^i}^2}} \tag{56}$$

$$\leq \frac{2\alpha}{(1-\beta_1)^2} \sqrt{\sum_{j=1}^{T} \|g_j^i\|_{x_j^i}^2}, \tag{57}$$

where the last inequality comes from lemma 8. $\qquad\square$

Putting everything together, we finally obtain

$$\sum_{t=1}^{T} f_t(x_t) - f_t(x_*) \leq \sum_{i=1}^{n} \sum_{t=1}^{T} \langle -g_t^i, \log_{x_t^i}(x_i^*) \rangle_{x_t^i} \tag{58}$$

$$\leq \frac{\sqrt{T} D_\infty^2}{2\alpha(1-\beta_1)} \sum_{i=1}^{n} \sqrt{v_T^i} + \frac{D_\infty^2}{2(1-\beta_1)} \sum_{i=1}^{n} \sum_{t=1}^{T} \beta_{1t} \frac{\sqrt{v_t^i}}{\alpha_t} \tag{59}$$

$$+ \frac{\alpha}{(1-\beta_1)^3} \sum_{i=1}^{n} (\zeta(\kappa_i, D_\infty) + 1) \sqrt{\sum_{t=1}^{T} \|g_t^i\|_{x_j^i}^2}, \tag{60}$$

where we used that for this choice of $\alpha_t$ and $\beta_{2t}$, we have $\alpha_t/\sqrt{v_t^i} \leq \alpha_{t-1}/\sqrt{v_{t-1}^i}$. Finally,

$$\frac{D_\infty^2}{2(1-\beta_1)} \sum_{i=1}^{n} \sum_{t=1}^{T} \beta_{1t} \frac{\sqrt{v_t^i}}{\alpha_t} \leq \frac{D_\infty^2 G_\infty n}{2\alpha(1-\beta_1)} \sum_{t=1}^{T} \sqrt{t}\beta_{1t} \leq \frac{\beta_1 D_\infty^2 G_\infty n}{2\alpha(1-\beta_1)(1-\lambda)^2}. \tag{61}$$

This combined with Eq. (53) yields the final result. $\qquad\square$

**Remark.** Notice the appearance of a factor $n/\alpha$ in the last term of the last equation. This term is missing in corollaries 1 and 2 of Reddi et al. (2018), which is a mistake. However, this dependence in $n$ is not too harmful here, since this term does not depend on $T$.

## C   AMSGRAD

**Theorem 5** (Convergence of AMSGRAD). *Let $(f_t)$ be a family of differentiable, convex functions from $\mathbb{R}^n$ to $\mathbb{R}$. Let $(x_t)$ and $(v_t)$ be the sequences obtained from Algorithm 1b, $\alpha_t = \alpha/\sqrt{t}$, $\beta_1 = \beta_{11}$, $\beta_{1t} \leq \beta_1$ for all $t \in [T]$ and $\gamma = \beta_1/\sqrt{\beta_2} < 1$. Assume that each $\mathcal{X}_i \subset \mathbb{R}$ has a diameter bounded by $D_\infty$ and that for all $1 \leq i \leq n$, $t \in [T]$ and $x \in \mathcal{X}$, $\|(\mathrm{grad} f_t(x))\|_\infty \leq G_\infty$. For $(x_t)$ generated using the AMSGRAD (Algorithm 1b), we have the following bound on the regret*

$$R_T \leq \frac{\sqrt{T} D_\infty^2}{2\alpha(1-\beta_1)} \sum_{i=1}^{n} \sqrt{\hat{v}_T^i} + \frac{D_\infty^2}{2(1-\beta_1)} \sum_{i=1}^{n} \sum_{t=1}^{T} \beta_{1t} \frac{\sqrt{\hat{v}_t^i}}{\alpha_t} +$$

$$\frac{\alpha\sqrt{1+\log T}}{(1-\beta_1)^2(1-\gamma)\sqrt{1-\beta_2}} \sum_{i=1}^{n} \sqrt{\sum_{t=1}^{T} (g_t^i)^2} \tag{62}$$

*Proof.* See Theorem 4 of Reddi et al. (2018). $\qquad\square$

# D    USEFUL LEMMAS

The following lemma is a user-friendly inequality developed by Zhang & Sra (2016) in order to prove convergence of gradient-based optimization algorithms, for geodesically convex functions, in Alexandrov spaces.

**Lemma 6** (Cosine inequality in Alexandrov spaces)**.** *If $a$, $b$, $c$, are the sides (i.e., side lengths) of a geodesic triangle in an Alexandrov space with curvature lower bounded by $\kappa$, and $A$ is the angle between sides $b$ and $c$, then*

$$a^2 \leq \frac{\sqrt{|\kappa|}c}{\tanh(\sqrt{|\kappa|}c)}b^2 + c^2 - 2bc\cos(A). \tag{63}$$

*Proof.* See section 3.1, lemma 6 of Zhang & Sra (2016).    □

**Lemma 7** (An analogue of Cauchy-Schwarz)**.** *For all $p, k \in \mathbb{N}^*$, $u_1, ..., u_k \in \mathbb{R}^p$, $a_1, ..., a_k \in \mathbb{R}_+$, we have*

$$\| \sum_i a_i u_i \|_2^2 \leq \left( \sum_i a_i \right) \left( \sum_i a_i \| u_i \|_2^2 \right). \tag{64}$$

*Proof.* The proof consists in applying Cauchy-Schwarz' inequality two times:

$$\| \sum_i a_i u_i \|_2^2 = \sum_{i,j} a_i a_j u_i^T u_j \tag{65}$$

$$= \sum_{i,j} \sqrt{a_i a_j} \left( \sqrt{a_i} u_i \right)^T \left( \sqrt{a_j} u_j \right) \tag{66}$$

$$\leq \sum_{i,j} \sqrt{a_i a_j} \| \sqrt{a_i} u_i \|_2 \| \sqrt{a_j} u_j \|_2 \tag{67}$$

$$= \left( \sum_i \sqrt{a_i} \| \sqrt{a_i} u_i \|_2 \right)^2 \tag{68}$$

$$\leq \left( \sum_i a_i \right) \left( \sum_i \alpha_i \| u_i \|_2^2 \right). \tag{69}$$

□

Finally, this last lemma is used by Reddi et al. (2018) in their convergence proof for ADAMNC. We need it too, in an analogue lemma.

**Lemma 8** ((Auer et al., 2002))**.** *For any non-negative real numbers $y_1, ..., y_t$, the following holds:*

$$\sum_{i=1}^{t} \frac{y_i}{\sqrt{\sum_{j=1}^{i} y_j}} \leq 2 \sqrt{\sum_{i=1}^{t} y_i}. \tag{70}$$

