# OpenReview forum: "Riemannian Adaptive Optimization Methods"
_ICLR.cc/2019/Conference_

### Official Review · AnonReviewer2 · 2018-10-25
**This paper is well-writen except a few flaws (see below). The proposed methods are potentially important in some applications. Therefore, I suggest publish this paper after addressing the comments below.**

**Rating:** 7
**Confidence:** 4

**Review:**

This paper presents Riemannian versions of adaptive optimization methods, including ADAGRAD, ADAM, AMSGRAD and ADAMNC. There are no natural coordinates on a manifold. Therefore, the authors resort to product of manifolds and view each manifold component as a coordinate. Convergence analyses for those methods are given. The the theoretical results and their Euclidean versions coincide. An experiment of embedding a tree-like graph into a Poincare model is used to show the performance of the Riemannian versions of the four methods.

This paper is well-written except a few flaws (see below). I do not have time to read the proofs carefully. The proposed methods are potentially important in some applications. Therefore, I suggest publish this paper after addressing the comments below.

Remarks:
*) P1, line 2: it particular -> in particular.
*) P3, line 9: Is R_x(v) = x + v most often chosen? A manifold is generally nonlinear. A simple addition would not give a point in the manifold.
*) P5, in Assumptions and notations paragraph: what are T and [T]? Is T the number of total iterations or the number of functions in the function family. The subscript of the function f_t seems to be an index of the functions. But its notation is also related to the number of iterations, see (8) and the algorithms in Figure 1.
*) P5, Figure 1: does a loop for the index $i$ missing?
*) Section 5: it would be clearer if the objective function is written as L:(D^n)^m \to R: \theta-> , where m is the number of nodes. Otherwise, it is not obvious to see the domain.
*) P7, last paragraph: Tables 2 and 3 -> Figures 2 and 3.
*) Besides the application in the experiments, it would be nice if more applications, at least references, are added.

---

> ### Author Response · Authors · 2018-11-19
> **Thank you for reviewing our work**
>
> Thank you for your detailed feedback. We have updated our paper according to your suggestions.
>
>
> We reply below to each of your remarks, more specifically.
>
> *) Typo corrected.
> *) Retraction: indeed, choosing the retraction R_x(v)=x+v requires having immersed the manifold into an ambient Euclidean space: note that we only say that the retraction is “most often chosen as” such, not that this choice is always a valid one. We mention it here because it is the one we used in our experiments.
> *) T is the number of iterations, [T] denotes the set of integers from 1 to T. We use same notations as in [1]: Each f_t is the objective function of the parameters to be optimized, evaluated at the batch taken at time t. For instance, when training a neural network, one could alternatively write f_t(x) = \sum_{y\in S_t} L(x,y), where x is the set of parameters of the model, L is the loss, each y is an input to the network, and S_t the (mini)batch taken at time t.
> *) yes, these are coordinate-wise operations. We did not write explicitly the loop over i to not influence the reader into implementing this algorithm with a loop over i. In most languages, such as python or C++, coordinate-wise operations such as adding vectors are highly optimized in the standard library. One could rewrite the algorithm without the “i”, with coordinate wise operations on vectors.
> *) We added a footnote clarifying the domain of the loss function in Section 5.
> *) Tables 2 & 3 -> figures 2 & 3: Thank you, we corrected this typo.
> *) Other potential applications include any optimization-based graph or word embedding method on a manifold. Note that the product-structure assumption is natural, since if one needs to embed n nodes into a manifold, the parameter space is a product of n manifolds.
> Following your suggestions, we have added a few references [2,3,4] as suggestions for further experiments, at the beginning of the experiment section.
>
>
> [1] On the convergence of Adam and beyond, Reddi et al., ICLR 2018
> [2] Representation trade-offs for hyperbolic embeddings, De Sa et al., ICML 2018
> [3] Hyperbolic entailment cones for learning hierarchical embeddings, Ganea et al., ICML 2018
> [4] Learning continuous hierarchies in the Lorentz model of hyperbolic geometry, Nickel & Kiela, ICML 2018

---

### Official Review · AnonReviewer1 · 2018-10-29
**Riemannian ADAM**

**Rating:** 7
**Confidence:** 5

**Review:**

I have enjoyed reading this paper. The paper is accessible in most cases and provides a novel optimization technique. Having said this, I have a few concerns here,


- I am not sure why the notion of product manifolds is required in developing the technique. To me, all the arguments follow without that. Even if the authors are only interested in manifolds that can be constructed in a product manner (say R^n from R),  the development can be done without explicitly going along that path. Nevertheless I may have missed something so please elaborate why product manifolds. I have to add that in many cases, the underlying Riemannian geometry cannot be derived as a product  space. For example, the SPD manifold cannot be constructed as a product space of lower dimensional geometries.

- I have a feeling that finding the operator \Pi in many interesting cases is not easy. Given the dependency of the developments on this operator, I am wondering if the method can be used to address problems on other manifolds such as SPD, Grassmannian or Stiefel. Please provide the form of this operator for the aforementioned manifolds and comment on how the method can be used if such an operator is not at our disposal.

- While I appreciate the experiments done in the paper,  common tests (e.g., Frechet means) are not presented in the paper (see my comment below as well).

- last but not least, the authors missed the work of   Roy et. al., "Geometry Aware Constrained Optimization Techniques for Deep Learning", CVPR'18 where RSGC with momentum and Riemannian version of RMSProp are developed. This reference should be considered and compared.


Aside from the above, please

- define v and \hat{v} for Eq.(5)

- provide a reference for the claim at l3-p4 (claim about the gradient and Hessian)

- maybe you want to mention that \alpha -> 0 for |g_t^i| at the bottom of p4

- what does [.] mean in the last step of the algorithm presented in p7

- what is the dimensionality of the Hn in the experiments

---

> ### Author Response · Authors · 2018-11-19
> **Thank you for reviewing our work**
>
> Thank you for your interest and professionalism. We reply below to each of your concerns.
>
> Product structure:
>
> (i) The product structure is natural for any optimization-based graph or word embedding method: if one wants to embed n nodes into a manifold M, then the parameter space is M^n. In particular, this would apply also if M is a PSD manifold.
>
> (ii) We noticed recently that other very recent approaches propose to also embed each point into a product of spaces, arguing that it allows the embeddings to benefit from the metric properties of each space [1,2].
>
> (iii) Our proof arguments would not hold without this product structure. This is easier to see from the convergence proof of Euclidean AMSgrad [3], appendix D, Eq.(18), where the last equality exploits the Euclidean coordinate system to expand the squared norms. This is not possible on a general Riemannian manifold. However, with a product structure, one can expand squared distances in the product manifold, as the sum of squared distances in each manifold of the product.
>
>
> Pi operator:
>
> -The presence of the projection operator is mostly useful for the convergence proof, to guarantee that the learning trajectory in parameter space is bounded, hence the presence of D_\infty in the bounds.
> -Note that this operator is also required to obtain theoretical bounds for Euclidean AMSgrad/Adam, even though it is often omitted in practice.
> -Note that in [4, section 3], it is assumed to be given, as a “projection oracle”. Also note that for many applications of interest, such as computing Karcher means on PSD manifolds (as done in [4, section 4]), a projection operator is not used nor needed for convergence, since the trajectory is trivially bounded (formally, this amounts to choosing a trivial projection into a ball containing the trajectory).
> -In the Poincaré ball, if X is a ball centered at the origin (as in our experiments), then the projection is naturally given by the parametrization in the Euclidean ambient space.
>
>
> Fréchet means:
>
> -This is an interesting suggestion that we will keep in mind for future work.
>
>
> Related work:
>
> -Thank you for pointing us to this relevant reference. We have added it to the related work section.
>
>
> Other remarks:
>
> In Eq.(5), \hat{v} is defined from v, which is defined as for Adam, defined just above. We have added a footnote explaining this.
> Added a reference for the claim line 3, page 4, about gradient and Hessian.
> At the bottom of page 4, \alpha is not required to go to 0: we consider here an (R)SGD update of fixed size \alpha.
> If by [.] you refer to gyr[. , .], this square bracket comes from the notation of the gyro-operator, for which we provide a reference. We use it in our experiments to efficiently compute parallel transport in the Poincaré ball.
> The dimension we use in our experiments is 5, as suggested in [5]. Added to experiments section.
>
>
> [1] Learning mixed curvature representations in product spaces,
> https://openreview.net/forum?id=HJxeWnCcF7
> [2] Poincaré Glove: hyperbolic word embeddings,
> https://openreview.net/forum?id=Ske5r3AqK7
> [3] On the convergence of Adam and beyond, Reddi et al., ICLR 2018
> https://openreview.net/forum?id=ryQu7f-RZ
> [4] First order methods for geodesically convex optimization, Zhang & Sra, JMLR 2016
> proceedings.mlr.press/v49/zhang16b.pdf
> [5] Poincaré embeddings for learning hierarchical representations, Nickel & Kiela, NIPS 2017
> https://arxiv.org/abs/1705.08039

---

### Official Review · AnonReviewer3 · 2018-11-05
**Riemannian Adam/Amsgrad on product manifolds with convergence guarantee but not supported well by experiments.**

**Rating:** 7
**Confidence:** 3

**Review:**

The paper extends Euclidean optimization methods, Adam/Amsgrad, to the Riemannian setting, and provides theoretical convergence analysis which includes the Euclidean versions as a special case. To avoid breaking the sparsity, coordinate-wise updates are performed on product manifolds.

The empirical performance seems not very good, compared to RSGD which is easier to use.

---

> ### Author Response · Authors · 2018-11-19
> **Thank you for reviewing our work**
>
> Thank you for reviewing our work. Even though RSGD is indeed slightly easier to use, we will make our code available to facilitate the use of our algorithms.

---

### Public Comment · (anonymous) · 2018-09-27
**Can you compute exponential map and parallel transport in general?**

A quick comment. Seems you use exponential map and parallel transport in your algorithm, and I guess it's okay to compute exponential map and parallel transport for some simple manifold. But is it usually possible for cases where people are interested? What I often see is retraction instead of exponential map, and sometimes parallel transport has no close form solution. It seems also unusual to assume cartesian product exists.

As you said, you probably can replace exp map with retraction, so it's better to stick to it, say in algorithm figure 1 and your proof.

For example section, people have some manifolds in mind, like sphere, orthogonal group/stiefel, grassmann, etc. so maybe it helps to address one of them.

---

> ### Public Comment · (anonymous) · 2018-09-28
> **Retraction on page 3**
>
> And for retraction, you cannot do R_x(v) = x+v since it maps to manifold. Maybe a projection.

---

> > ### Public Comment · (anonymous) · 2018-09-29
> > **A matter of notations.**
> >
> > Concerning the use of a projection operation, note that we include it both in our algorithms and convergence proofs. As soon as you have an extrinsic representation of your manifold in an ambient Euclidean space (which we do in our experimental setup), the + operation is well defined, and whether you include the projection in the retraction or apply it on top is just a matter of notations. See [5] for similar definitions/notations as ours.
> >
> > [5] Poincare embeddings for learning hierarchical representations, Nickel & Kiela, NIPS 2017

---

> ### Public Comment · (anonymous) · 2018-09-29
> **Thanks for you interest!**
>
> Hi, thank you for your interest!
>
> 1. Although the exponential map is not always known in closed-form, your remark also applies to any Riemannian optimization algorithm, even to Riemannian SGD. Note however that its formula is known for spherical and hyperbolic spaces, Stiefel and Grassmann manifolds, as well as for a variety of matrix Lie groups, which correspond nowadays to the main application cases of Riemannian optimization. In cases where it is not, a retraction mapping can be used instead, but the choice of your retraction will affect the trajectory. Although comparing the use of various retractions could constitute interesting future work, this was not our theoretical focus.
>
> 2. Could you please share with us why you think it would be better to stick with the retraction mapping in general? Besides the fact that using the exp map is more mathematically principled, note that [1] also obtained significantly better empirical results by using fully Riemannian methods in the Lorentz model of hyperbolic geometry. In practice, what is best seems to depend on the manifold, the model and the task.
>
> 3. As explained in the last paragraph of our introduction, the product structure is relevant to any method aiming to embed a set of points in a Riemannian manifold M. This concerns in particular all optimization-based graph and word embedding methods. If k is the size of the set of vertices or words to embed, then the parameter space is M^k. Moreover, note that one could also choose M itself as a product of manifolds, as was done for instance in [2,3,4].
>
> 4. Concerning the closed-form formula of parallel-transport, although it is also given in the above mentioned cases, it can be seen from our proofs that our convergence theorems still hold if it is replaced by any isometry between the corresponding tangent spaces. We will add this as an interesting remark, thank you for pointing it out! Also notice that in our experiments section, we provide the formula for parallel-transport in the Poincare ball, using gyro operations.
>
>
> [1] Learning Continuous Hierarchies in the Lorentz Model of Hyperbolic Geometry, Nickel & Kiela, ICML 2018
> [2] https://openreview.net/forum?id=HJxeWnCcF7
> [3] https://openreview.net/forum?id=Ske5r3AqK7
> [4] https://openreview.net/forum?id=r1xRW3A9YX

---

> > ### Public Comment · (anonymous) · 2018-09-29
> > **Thanks for your response**
> >
> > Thanks for your response.
> >
> > For 1. and 2., I just think if you claim that the convergence result with retraction is not too different, the statement is more general since exp map is also a retraction, see https://arxiv.org/abs/1802.09128. I agree that for those manifolds exp map is easy to compute, retraction is an approximation of exp map and for general manifolds it is easier. If the convergence guarantee is different for retraction and exp map, it may be also worth pointing out and comparing.
> >
> > For 3., the product structure is not used for sphere, etc., so it's just unusual to me. It helps reader if you stress it, and recall some basics or intuitions, such as Sec 3.2 (d(x,y))^2 = \sum d(x^i,y^i)^2. Maybe you can add a few line proof in appendix or just quote some literature.
> >
> > For 4. is there a closed form solution for parallel translation for Stiefel manifold (see ex 8.1.2 Optimization Algorithms on Matrix Manifolds by Absil et al)? I'm not so sure about the isometry stuff, maybe you can explain more, and discuss how such "inexact" parallel translation interacts with geodesic convexity, Lipschitz constants and so on. But anyways, you have a concrete example where the algorithm works, so the statement are no doubt legit.
> >
> > And I see your point for defining retraction in another reply.
> >
> > Thanks again for your response! And my apology it's a quick comment so I did not go through every detail.

---

### Author Response · Authors · 2018-11-19
**Many thanks for reviewing our paper.**

We would like to thank all three reviewers for their work and their interest in our work.

We have incorporated the modifications suggested by reviewers and are open to further comments.

Please find more detailed responses below each review.

---

### Public Comment · (anonymous) · 2018-11-28
**Issue with any isometry**

You claim your regret bound holds for any isometry phi. In the simple euclidean case that means that if for example you use phi(m) = -m your regret bound should still hold even though the optimization should fail. I even tried a simple script just to be sure and it indeed doesn't improve (just sanity check not claiming to check thoroughly). The gradient norm also do not increase so it is unlikely the bound holds in a vacuous way.
This might indicate a issue with the proof.

---

> ### Author Response · Authors · 2018-11-28
> **Many thanks for your comment**
>
> Our bounds indeed hold for any isometry phi.
> Even with phi(m) = -m.
> We understand your confusion.
>
> First, please note that if phi(m) = -m, although the past gradients in the exponential moving average would be alternated, i.e. multiplied by some (-1)^t, the current gradient is not modified by phi.
>
> That being said, what you are pointing out (“on the choice of phi”, bottom of page 6) shows that in the Euclidean case, if you would take phi to be a random orthogonal transformation, even a different one at each step, the bounds still hold.
>
> Isn’t this puzzling?
>
> Our Riemannian generalization sheds light on this part of the convergence proof of AmsGrad. Indeed, this can be easily seen from our proof, Eq.(18) and Eq.(30), that one only needs to know the “size” of the update “transported” by phi, and not its direction.
>
> It does not tell that the bounds of Reddi et al. are wrong, it just means that the current bounds that the community has on Adam-like algorithms are “too loose”.
>
> More precisely, although the currently known regret bounds guarantee convergence, they do not exploit acceleration coming from momentum. This suggests that much better bounds could potentially be obtained.
>
> We hope that this work will motivate further research in better understanding Adam-like algorithms.
>
> Thank you for your interest!

---

### Public Comment · ~Maxim_Vadimovich_Kochurov1 · 2018-12-28
**Unofficial implementation:)**

Hi! Thank you for this inspiring work! As soon as I saw the paper submitted I tried to implement it just for fun and use it for some areas in my research. It turned into a python package for pytorch. I've implemented Stiefel manifolds, Adam, Amsgrad and SGD so far due to limited time. A friend of mine helped me with adding some experimental samplers on Riemannian manifolds. Feel free to check it out on github/pip

https://github.com/ferrine/geoopt
https://pypi.org/project/geoopt/

---

> ### Author Response · Authors · 2019-01-31
> **Thanks a lot for sharing!**
>
> Very happy to hear!
>
> If you get new empirical insights on what is best to use, or applications where you found it useful, please feel free to post it on this thread :)
>
> Will keep an eye on your work!

---

> > ### Public Comment · ~Maxim_Vadimovich_Kochurov1 · 2019-05-04
> > **Adam is fine**
> >
> > Based on my and my colleague's experience, we found R-Adam easy to work with. Specifically, we tried to replicate https://papers.nips.cc/paper/7780-hyperbolic-neural-networks with changed optimizer (repo: [1]). Probably due to carefully tuned R-SGD in this work we were not able to reproduce the results with SGD. We used batch_size=1000 not to wait for convergence forever and that might affect SGD and be the reason for our failure to reproduce the results here.  On the contrary, increased batch_size does not affect R-Adam (as expected) and we get nice convergence with large batches. Empirically, we found that optimization is more smooth with smaller momentum betas (due to large batch_size?), we used (0.7, 0.9). Finally, we obtained similarly looking plots that compare Euclidean network to Hyperbolic one using Adam.
> >
> > [1] https://github.com/ferrine/hyrnn This repo needs to be polished a lot, we plan to move it into https://github.com/geoopt/ once we have a bit more free time to work on it.

---

### Meta-Review · Area_Chair1 · 2018-12-13
**An interesting analysis and algorithm**

**Confidence:** 4
**Recommendation:** Accept (Poster)

**Metareview:**

Dear authors,

All reviewers agreed that your work sheds new light on a popular class of algorithms and should thus be presented at ICLR.

Please make sure to implement all their comments in the final version.